# Sparse MoE as the New Dropout: Scaling Dense and Self-Slimmable Transformers

**Tianlong Chen[1]\*, Zhenyu Zhang[1]\*, Ajay Jaiswal[1], Shiwei Liu[1], Zhangyang Wang[1]**
[1]VITA Group, University of Texas at Austin
{tianlong.chen,zhenyu.zhang,ajayjaiswal,shiwei.liu,atlaswang}@utexas.edu

## Abstract

Despite their remarkable achievement, gigantic transformers encounter significant drawbacks, including exorbitant computational and memory footprints during training, as well as severe collapse evidenced by a high degree of parameter redundancy. Sparsely-activated Mixture-of-Experts (SMoEs) have shown promise to mitigate the issue of training efficiency, yet they are prone to (1) *redundant experts* due to representational collapse; and (2) *poor expert scalability for inference and downstream fine-tuning*, primarily due to overfitting of the learned routing policy to the number of activated experts during training. As recent research efforts are predominantly focused on improving routing policies to encourage expert specializations, this work focuses on *exploring the overlooked scalability bottleneck of SMoEs* and leveraging it to effectively **scale dense transformers**. To this end, we propose a new plug-and-play training framework, **SMoE-Dropout**, to enable scaling transformers to better accuracy in their full capacity without collapse. Specifically, SMoE-Dropout consists of a *randomly initialized and fixed* router network to activate experts and gradually increases the activated expert number as training progresses over time. Transformers trained by SMoE-Dropout naturally exhibit a **"self-slimmable"** property subject to resource availability, offering smooth and consistent performance boosts with an increase in activated experts during inference or fine-tuning. Our extensive experiments across diverse transformer architectures on a variety of tasks demonstrate the superior performance and substantial computation savings of SMoE-Dropout, compared to dense training baselines with equivalent parameter counts. In particular, our trained BERT outperforms its densely trained counterpart with consistent improvements of {1.03%, 0.78%, 1.09%} on challenging reasoning tasks {ASDiv-A, MAWPS, SVAMP}, respectively. Codes and models are available in https://github.com/VITA-Group/Random-MoE-as-Dropout.

## 1 Introduction

Scaling neural networks, historically with the blessing of modern hardware, have dramatically improved the state-of-the-art on a wide array of real-world machine learning applications and leaderboards, conforming to the empirical scaling laws (Kaplan et al., 2020), where the final model quality has been found to have a power-law relationship with the amount of data, model size, and compute time. Transformers (Vaswani et al., 2017), swiftly after their introduction, have become *de facto* choice for many natural language processing (NLP) (Yang et al., 2019c; Liu et al., 2019b; Talmor et al., 2018; Jaiswal et al., 2021; Yang et al., 2019b; Wang et al., 2018; Ding et al., 2019; Chowdhery et al., 2022; Wei et al., 2022) and computer vision (Dosovitskiy et al., 2020; Han et al., 2020; Touvron et al., 2021; Mao et al., 2022; Zheng et al., 2021; Parmar et al., 2018) applications and now their parameter counts are typically measured in billions rather than millions. Unfortunately, this *exploitation of parameters actuates a roughly quadratic blow-up in training costs*, as both the model size and the number of training examples increase especially for dense advanced transformer-based models (*e.g.,* BERT (Devlin et al., 2018) and GPT (Brown et al., 2020)) and require thousands of GPU days for training. Additionally, these gigantic transformers suffer from the **representation collapse** issue during vanilla training, which is affirmed by a high degree of parameter redundancy (Guo et al., 2019; Ganesh et al., 2020; McCarley et al., 2019) and observed ineffective usage of the transformer expressiveness (Michel et al., 2019; Chen et al., 2022a).

---

\*Equal Contribution.

Sparse Mixture-of-Experts (SMoEs) enable efficient scaling of model capacity at a fixed computational cost by performing *input-dependent conditional computing*. Such property facilitates training transformers with significantly high parameter counts at moderately increased cost, compared to their dense counterparts, resulting in *improved training efficiency*. For instance, with similar training FLOPS, Switch-Large (Fedus et al., 2021) (a kind of SMoE) is $35\times$ larger than a T5-Large dense model (Raffel et al., 2020). Despite their advantages in mitigating computational and energy footprints, SMoEs have many critical limitations. **Firstly**, the current learning-based routing mechanisms in SMoEs tend to push hidden representations clustering around expert centroids (Chi et al., 2022), implying a trend toward **representation collapse**, which in turn leads to redundant experts, inferior expert specialization, thereby substandard performance (Mittal et al., 2022; Chen et al., 2022b).

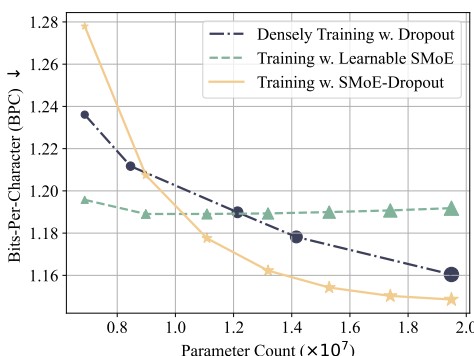

Figure 1: Bits-Per-Character ($\downarrow$) on `enwik8`'s test-set with a 4-layer Transformer-XL. SMoE-Dropout demonstrates a "self-slimmable" property where inference performance is smoothly boosted along with the increase of activated parameters. Learnable SMoEs tend to overfit certain levels of network capacity. Note that only gray curve is produced by (5) different dense models.

**Secondly**, SMoEs suffer from **poor scalability during inference and downstream fine-tuning** prominently due to *overfitting of the learned routing policy* to the number of activated experts during training. Naive solutions to mitigate such sparsity immutability often lead to performance degradation. As recent research efforts for SMoEs are predominantly focused on improving routing policies to encourage expert specializations, we explore the overlooked scalability bottleneck of SMoEs and ask: *Does there exist a principled and pluggable approach to modify SMoE training that can enhance scalability at inference and downstream fine-tuning of large-scale transformers, by dynamically adapting the number of activated experts subject to resource availability?*

To this end, this paper proposes a novel plug-and-play training framework, named **SMoE-Dropout**, to enable scaling transformers to better accuracy in the full capacity setting without collapse. More specifically, SMoE-Dropout employs a fixed router network that is randomly initialized to activate experts and progressively increases their number as training progresses over time. Our simple, yet highly effective strategy has a multi-fold win-win for trained transformers, specifically: ❶ obtaining a **"self-slimmable"** property during inference and downstream fine-tuning subject to resource availability, which delivers a once-for-all in-situ trade-off between efficiency and performance; ❷ mitigating *representational collapse* and effectively utilizing the full model capacity, where activating more experts produces superior performance (Figure 1 (blue)); ❸ eliminating the overhead of learning routing policies for SMoE. Note that SMoE-Dropout can be swiftly adapted for training any deep learning network (e.g. CNNs), given some splitting techniques (Zhang et al., 2021), but this work primarily focuses on transformers considering their exploding computational footprints. Our innovative contributions can be summarized as:

⋆ We propose a new plug-and-play training framework, **SMoE-Dropout**, to enable scaling transformers in the full capacity setting without collapse. SMoE-Dropout facilitates the randomly and sparsely activated structure of network modules, playing an *implicit regularization role similar to dropout*. Our new framework leads to enhanced generalization and reduced training costs (*e.g.*, up to $37\%$ running time savings) compared to the vanilla training of large dense transformers at equivalent parameter counts.

⋆ Transformers trained by SMoE-Dropout naturally exhibit a **"self-slimmable"** property that displays smooth and consistent performance boosts when increasing activated experts during inference or fine-tuning (Figure 1 (blue)). This property enjoys an "in-situ" trade-off between efficiency and performance at deployment, subject to resource availability.

⋆ Our extensive experiments across representative architectures on a variety of tasks validate the effectiveness of our proposed SMoE-Dropout. Specifically, during pre-training, our approach has $\{1.37, 4.10\}$, $\{2.53, 12.44\}$ and $\{154.12, 188.00\}$ $(\times 10^{-2})$ lower BPC than {vanilla dense training (with the same parameter counts), learned SMoE} for Transformer-XL, BERT, and RoBERTa, respectively; after transferring, SMoE-Dropout obtains $\{0.07\%, 1.03\%, 0.78\%, 1.09\%\}$ performance improvements for BERT and $\{-, 5.88\%, 0.07\%, 5.04\%\}$ for RoBERTa, on {CSQA, ASDiv-A, MAWPS, SVAMP} reasoning tasks compared to its dense training counterpart.

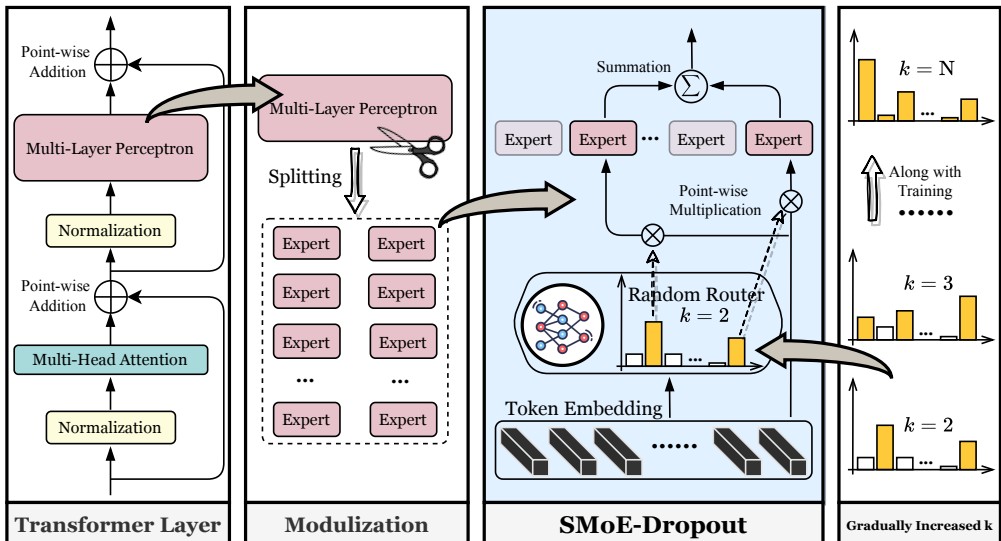

Figure 2: Overview of our proposed SMoE-Dropout. *Left* describes the standard transformer layer, consisting of multi-head attention and multi-layer perceptron (MLP) components. *Middle Left* shows the process of modulization. It splits the original MLP evenly and constructs a series of experts which are smaller MLPs with a reduced hidden dimension. *Middle Right* presents the overall procedure of SMoE-Dropout. The random router selects the top-$k$ experts given a token embedding and then reweights the features from activated experts. In the end, a summation is conducted to aggregate all features. *Right* displays the gradually increased number of chosen experts, along with the training procedure.

## 2 RELATED WORKS

**Mixture of Experts (MoE).**  MoE is a special kind of neural network, where its parameters are partitioned into a series of sub-modules (commonly referred to as experts), and conditional computation is then performed in an input-dependent fashion (Jacobs et al., 1991; Jordan & Jacobs, 1994; Chen et al., 1999; Yuksel et al., 2012). The traditional dense MoEs are computationally intensive, as they adopt all experts for each input (Eigen et al., 2013). Fortunately, recent investigations (Shazeer et al., 2017; Lepikhin et al., 2020; Fedus et al., 2021) have proved the effectiveness of MoEs with sparsely activated experts (*i.e.*, SMoE) at both training and inference stages, which greatly trim down the cost and scale language models to enormous sizes like trillions of parameters (Fedus et al., 2021). This efficient fashion of SMoEs gains increasing popularity in various NLP (Shazeer et al., 2017; Lepikhin et al., 2020; Zhou et al., 2022; Zhang et al., 2021; Zuo et al., 2022; Jiang et al., 2021) and vision (Riquelme et al., 2021; Eigen et al., 2013; Ahmed et al., 2016; Gross et al., 2017; Wang et al., 2020; Yang et al., 2019a; Abbas & Andreopoulos, 2020; Pavlitskaya et al., 2020) tasks.

However, its sparse-gated manner incurs several downsides, including: (1) *Unstable training*. Zoph et al. (2022) pointed out that while techniques like gradient clipping can stabilize SMoE training, they often result in lower quality. The router $z$-loss (Zoph et al., 2022) is a preferred solution for achieving both improved performance and stability. (2) *Poor specialization*. One of the intriguing goals of SMoE is to divide-and-conquer the learning task by solving each piece of the task with adaptively selected experts (Aoki et al., 2021; Hazimeh et al., 2021; Ma et al., 2018; Mittal et al., 2022). To encourage specialization and decrease redundancy among experts (Chen et al., 2022b), Dai et al. (2022) pre-defined the expert assignment for different input categories, while Hazimeh et al. (2021) advocated multiple, diverse router policies. (3) *Representation collapse and load imbalance among experts*. As the primary issue of learning-based SMoEs, various approaches have been proposed to mitigate their negative effects. Shazeer et al. (2017) injected Gaussian noises into gating networks to promote the routing balance. Later, Lepikhin et al. (2020); Fedus et al. (2021) applied an auxiliary loss of load balancing regularizers; Lewis et al. (2021) performed the routing by dealing with a linear assignment problem; Clark et al. (2022) utilized reinforcement learners; Zhou et al. (2022) routed top-$k$ inputs per expert instead of selecting top experts per input sample. Beyond learned routing policies, Roller et al. (2021) and Zuo et al. (2022) designed deterministic hashing and stochastic assignments, respectively, which eliminate the necessity for router networks.

Zuo et al. (2022), one closely related prior work, also endorsed the advantage of stochastic expert assignment. They randomly activate experts for each input during training and inference, which leads

to inconsistent inference results. To address the prediction randomness, Zuo et al. (2022) employed a consistent regularized loss to penalize the discrepancy among different experts. However, such regularization is prone to redundancy in SMoEs and sacrifices the network capacity. In our proposal, the fixed router with random weights generates deterministic inferences. Meanwhile, the presented "self-slimmable" attribute suggests the full models' expressiveness is adequately exploited.

**Dropout and Other Training Techniques for Transformers in NLP.** Dropout (Srivastava et al., 2014) was developed to prevent overfitting in over-parameterized networks during training, by randomly omitting neurons and their corresponding connections. Follow-up studies develop plenty of dropout variants (Zhang & He, 2020; Wan et al., 2013; Ba & Frey, 2013; Kingma et al., 2015; Gal et al., 2017; Wu & Gu, 2015; Tompson et al., 2015; DeVries & Taylor, 2017; Park & Kwak, 2016; Semeniuta et al., 2016). In parallel, McAllester (2013); Mou et al. (2018); Mianjy & Arora (2020); Zhang & Xu (2022); Neklyudov et al. (2017); Gal & Ghahramani (2016) have devoted themselves in deriving the theoretical foundation for dropout and explaining its implicit regularization impacts.

Other notorious bottlenecks of transformer training primarily stem from overfitting and instability caused by poor optimization (Zhang et al., 2020; Liu et al., 2019a; 2020a), insufficient or heterogeneous downstream data (Variš & Bojar, 2021; Zhang & Vaidya, 2021), *etc.*. Accordingly, numerous remedies are developed to address the issues. For example, data augmentations (Sun et al., 2020), improved initialization (Liu et al., 2020b;a; Xu et al., 2020; Zhu et al., 2021), upgraded normalization (Wang et al., 2022; Yang et al., 2022), enhanced optimizers (Cohen et al., 2022), weight decay (Loshchilov & Hutter, 2017), and early stopping.

## 3 METHODOLOGY

### 3.1 PRELIMINARY

**Sparse Mixture of Experts (SMoEs).** SMoE models leverage conditional computation to activate different subsets of a network for different inputs. A building block of SMoEs is the expert layer including a multi-head attention block and multiple experts in parallel. In this work, we consider SMoE for Transformers, where SMoE layers are incorporated into contiguous Transformer blocks. SMoE expert can be normally constructed by either splitting the vanilla MLP of transformers into smaller pieces (Zhang et al., 2021) or replicating the MLP (Fedus et al., 2021). Most existing SMoE works mainly concentrate on the MLP component in transformers since MLPs constitute roughly $2/3$ of total model parameters counts storing substantial amounts of learned knowledge as memory networks (Geva et al., 2020; Dai et al., 2021).

Let $\{\mathcal{E}_i\}_{i=1}^{N}$ denote the experts, where $i$ is the index of expert and N is the total number of experts. A gating network or router $\mathcal{R}$ is inserted to choose the top-$k$ experts with the largest scores $\mathcal{R}(\boldsymbol{x})_i$, and $\boldsymbol{x}$ represents the input embedding. Usually, $k \ll N$, which implies a sparsely activated setting. Specifically, the resultant output of the expert layer can be depicted as follows:

$$\boldsymbol{y} = \sum_{j=1}^{k} \mathcal{R}(\boldsymbol{x})_j \cdot \mathcal{E}_j(\boldsymbol{x}); \mathcal{R}(\boldsymbol{x}) = \texttt{TopK}(\texttt{softmax}(\mathcal{G}(\boldsymbol{x})), k); \texttt{TopK}(\boldsymbol{v}, k) = \begin{cases} \boldsymbol{v} & \text{if } \boldsymbol{v} \text{ is the top } k \\ 0 & \text{otherwise} \end{cases} \quad (1)$$

where $\mathcal{G}$ is the critical part of a router $\mathcal{R}$. For a learnable routing, $\mathcal{G}$ is a neural network that can be one or a few layers MLP (Shazeer et al., 2017; Fedus et al., 2021). $\mathcal{E}_j(\boldsymbol{x})$ stands for features from the expert $\mathcal{E}_j$. It will be further summed with a scaling coefficient $\mathcal{R}(\boldsymbol{x})_j$ to form the final output $\boldsymbol{y}$. The $\texttt{TopK}$ function maintains the largest $k$ values and sets the reset elements to zero. In practice, a load or important balancing loss (Shazeer et al., 2017) is employed to avoid the representation collapse issue, *i.e.*, always picking the same experts for different inputs and ignoring others.

**Dropout and its variants.** Dropout is a conventional training technique employed to alleviate the risk of overfitting. The vanilla dropout is typically applied to fully connected layers with a dropping probability $p$. During each training iteration, neurons will be disabled with the probability $p$. In other words, the omission of neurons follows a $\texttt{Bernoulli}(p)$ distribution. As for the inference phase, there is no dropout and all neurons are activated. To counterbalance the surplus information during training, the output logits are reweighted by $1 - p$. In this paper, we selected two representatives among diverse proposed dropout variants, concrete dropout (Gal et al., 2017) and dropblock (Ghiasi et al., 2018) as our comparison baselines.

▷ *Concrete Dropout.* It replaces the discrete $\texttt{Bernoulli}(p)$ distribution of dropout with a continuous relaxation, *i.e.*, $\texttt{Concrete}$ distribution, and allows an automatic tuning of the dropping probability $p$. For example, considering the one-dimensional case, as shown in Gal et al. (2017), a

`Concrete` random variable $z$ is described as $z = \texttt{sigmoid}\left(\frac{1}{t} \times \left(\log(p) - \log(1-p) + \log(u) - \log(1-u)\right)\right)$, where $u \sim \texttt{Unif}(0,1)$ is a uniform random variable and $t$ denotes a temperature hyperparameter. Note that parameter $p$ is optimized in a data-driven way.

▷ *DropBlock.* Instead of performing Bernoulli dropping per feature map, Ghiasi et al. (2018) applies it in areas within feature maps. They claim that DropBlock improves the generalization and limits overfitting by hiding certain areas of features or input samples.

## 3.2 A New Training Pipeline: SMoE-Dropout

**Modulization.** The first step in our SMoE-Dropout, turns a large densely connected MLP into multiple smaller MLPs with the same size, as demonstrated in Figure 2. Without loss of generality, in Figure 2, we use a single-layer MLP $f$ with a dimension $d$ for illustrations. After the modulization, it is divided into a set of MLPs $\{\mathcal{E}_1, \mathcal{E}_2 \cdots, \mathcal{E}_N\}$, where they have the same hidden dimension $\frac{d}{N}$.

**Random Routing Policy.** Few prior works have investigated some form of random routing policies, such as Roller et al. (2021) utilizes a hash table to enforce a pre-defined deterministic random mapping from inputs to experts and Zuo et al. (2022) adopts a fully random assignment in each training iteration. Although they have shown some benefits from random policies, both methods suffer from inconsistent inference predictions, and can not outperform the densely trained models with equivalent parameter counts. In contrast, our proposed framework, SMoE-Dropout considers a *randomly initialized and fixed router network* to guide token assignment. Different from previous works, our proposal's assignment is (1) implicitly optimized during training, since feature embeddings remain updated for the same input sample; (2) deterministic during inference thanks to the fixed weights in $\mathcal{R}$. Extensive results in Section 4 verify the superiority of our proposal, compared to existing random policies and the dense baseline with the same model parameters.

Additionally, another crucial design in SMoE-Dropout's routing is the *progressively enlarged number of activated experts (k)*. Riquelme et al. (2021); Jiang et al. (2021) reveal that altering $k$ in the inference phase incurs significant performance degradation if the SMoE is learned with a fixed $k$. For example, (Riquelme et al., 2021)'s SMoE trained with $k = 1$ has $20\% \sim 30\%$ accuracy drops on ImageNet, when activating $k \geq 7$ experts during the evaluation. This drawback substantially restricts the practical use of SMoEs because diverse real-world scenarios require different resource budgets, necessitating flexible and effective network capacity during inference. To tackle this limitation, we adopt a training strategy that gradually enriches the active network capacity by linearly increasing the number of selected experts $k$ during training. This approach coincides with the principle of curriculum learning and provides the attractive **"self-slimmable"** ability, which *consistently boosts performance* for transformers as the number of activated experts increases during inference and downstream fine-tuning, as shown in Figure 1.

**SMoE-Dropout.** Our effective proposal comprises three simple and highly effective steps, as described in Figure 2. First, it divides the MLP into a series of MLPs with a reduced size for modulization (*Middle Left* of Figure 2). Then, a random policy parameterized by fixed weights is introduced to route token embeddings to $k$ experts with the largest response (*Middle Right* of Figure 2). Finally, it progressively actives more experts, preventing the overfitting to the amounts of used network capacity during training. (*Right* of Figure 2).

## 4 Experiment

### 4.1 Implementation Details

**Network Architectures and Comparison Baselines.** In our experiments, we have adopted three representative transformer-based networks, including BERT (Devlin et al., 2018), Transformer-XL (Dai et al., 2019), and RoBERTa (Liu et al., 2019b). Specifically, we use double-size BERT$_{\text{base}}$ / RoBERTa$_{\text{base}}$ that have 12 transformer layers, 768-dimensional encoder layers, 6144-/3072-dimensional feed-forward networks (MLPs), and 12 attention heads. For both Transformer-XL, we choose a reduced size due to limited resources, which has 4 layers, 256-dimensional encoder layers, 8192-dimensional feed-forward networks, and 8 attention heads with a head size of 64.

For sufficient comparisons with our proposal, Training w. SMoE-Dropout, we consider five baselines: $(i)$ Densely Training w. Dropout, where the vanilla dropout is applied to feed-forward networks (MLPs); $(ii)$ Densely Training w. Concrete Dropout (Gal et al., 2017); $(iii)$ Densely Training w. DropBlock (Ghiasi et al., 2018). Note that both Concrete Dropout and DropBlock are inserted in feed-forward networks, replacing the vanilla dropout; $(iv)$ Training w. Learnable SMoE (Fedus et al., 2021); $(v)$ Training w. THOR (Zuo et al., 2022), where THOR is another random SMoE

that randomly activates a pair of experts for each input sample and adopts an auxiliary consistency regularization based on Kullback-Leibler (KL) divergence. To compute the regularization term, two forward processes are needed in each training iteration.

**Pre-Training.** ▷ *Datasets.* Transformer-XL is pre-trained on `enwik8` (Mahoney, 2011) dataset, while we use `BooksCorpus` (Zhu et al., 2015) for BERT and RoBERTa. ▷ *Training Configurations.* For Transformer-XL, we follow the official training setups, using Adam optimizer and the learning rate starts from $2.5 \times 10^{-4}$ and decreases according to a cosine annealing scheduler. We use a batch size of 22 and optimize the network for $4 \times 10^5$ iterations. As for BERT pre-training, we adopt an AdamW optimizer with an initial learning rate of $5 \times 10^{-5}$ that linearly decays to 0. The batch size and total training steps are 64 and $1 \times 10^5$, respectively. RoBERTa's pre-training configurations strictly follow the default from HuggingFace[1], but with reduced training steps of $1 \times 10^5$. Moreover, we conduct a grid search and set the coefficient of THOR's regularization term as 2. Similarly, the temperature in Concrete dropout is $t = 0.1$. ▷ *Evaluation Metrics.* Since both performance and efficiency are essential, we assess the pre-training performance via Bits-Per-Character (BPC) on the hold-out validation set, where a smaller BPC value indicates a better pre-training; and we report training time per iteration & the number of floating point operations (FLOPs) of single-sample inference, for evaluating the efficiency. {1 RTX A6000, batch size 22} and {8 V100, batch size 64} are adopted for time measurements of Transformer-XL and BERT/RoBERTa, respectively.

**Downstream Fine-Tuning.** ▷ *Datasets.* Five benchmarks across three downstream tasks are examined in this paper, including text classification (`SST-2` (Socher et al., 2013)), arithmetic reasoning (`ASDiv-A` (Miao et al., 2020), `MAWPS` (Koncel-Kedziorski et al., 2016), `SVAMP` Patel et al. (2021)), and commonsense reasoning (`CSQA` (Talmor et al., 2018)). ▷ *Training Configurations.* We perform dense fine-tuning for all approaches. Given a downstream parameter budget, SMoE-based methods will select the most voted experts based on their routing policies. Detailed training setups are listed as follows. We fine-tune the pre-trained Transformer-XL with a smaller learning rate of $1 \times 10^{-4}$ and a batch size of 64 on `SST-2` benchmark. And for BERT and RoBERTa, we fine-tune the models on the aforementioned four reasoning datasets. The learning rate is fixed at $2 \times 10^{-5}$ and the batch size is 64. In each downstream task, the fine-tuning continues for 3 epochs, while other configurations are kept the same as the ones in pre-training. ▷ *Evaluation Metrics.* At the evaluation phase, accuracy (%) and the problem solving rate (%) (Wei et al., 2022) are reported on the test set of `SST-2` and other reasoning tasks, respectively.

### 4.2 SUPERIOR PERFORMANCE OF SMoE-DROPOUT

We adopt classical transformer-based models, *i.e.*, {Transformer-XL, BERT, RoBERTa}, and train them in a dense or SMoE-based manner on {`enwik8`, `BookCorpus`, `BookCorpus`}. Evaluation results are summarized in Table 1, where all models are compared under the same number of parameter counts. The following observations can be drawn: ❶ Our SMoE-Dropout demonstrates superior performance compared to all other training algorithms. Specifically, SMoE-Dropout with all experts selected obtains $1.37 \sim 18.49$, $0.56 \sim 12.44$, and $152.82 \sim 188.00$ $(\times 10^{-2})$ lower BPC for Transformer-XL, BERT, and RoBERTa, respectively. This validates the effectiveness of our proposals. ❷ Appropriate random routing policies show consistent performance benefits across all three network backbones. Moreover, our randomly weighted router surpasses the completely random allocation in THOR, which is within expectation since our assignment is implicitly "optimized" using evolved feature embeddings. ❸ In terms of training efficiency, SMoE-Dropout has up to 21%, 37%, and 25% training time savings compared to the dense training of three backbones. If only half of the experts ($k = \frac{N}{2}$) are activated, our approach enjoys extra $23\% \sim 34\%$ inference FLOPs reduction with a comparable BPC. Although the learnable SMoE reaches the best efficiency, it results in inferior performance.

Besides, we report another group of experiments varying the expert numbers (parameter counts) during evaluation. As shown in Figure 3, for SMoE-based approaches, we directly change the number of activated experts at the inference stage, which is an *in-situ* fashion from the single trained transformer. While for dense training baselines, each dot in their curve requires a separately trained model since it does not allow modifications of network capacity without further fine-tuning. Our findings are as follows: ❶ The performance of SMoE-Dropout is stably improved along with more parameters used, and it outperforms the others after 1.0, 10, and 8 $(\times 10^7)$ parameter counts for three backbones. Such "slimmable" property enables scaling transformers to the full capacity without

---

[1] https://huggingface.co/docs/transformers/model_doc/roberta.

Table 1: Testing performance of {Transformer-XL, BERT, RoBERTa} network backbones on {`enwik8`, `BookCorpus`, `BookCorpus`} datasets, respectively. **All models are compared under the same number of parameter counts**. Training time (s) and inference FLOPs ($\times 10^{10}$) are reported. For THOR (Zuo et al., 2022), SMoE, and SMoE-Dropout, evaluations are performed with half ($k = \frac{N}{2}$) or all ($k = N$) experts activated.

| Methods | Transformer-XL | | | BERT | | | RoBERTa | | |
|---|---|---|---|---|---|---|---|---|---|
| | BPC ($\downarrow$) | Time | Infer. FLOPs | BPC ($\downarrow$) | Time | Infer. FLOPs | BPC ($\downarrow$) | Time | Infer. FLOPs |
| Dense w. Dropout | 1.1623 | 5.1298 | 7.7579 | 7.6546 | 0.2088 | 135.72 | 8.0903 | 0.1898 | 101.75 |
| Dense w. Concrete Dropout | 1.3335 | 6.3519 | 7.7579 | 7.6419 | 0.3031 | 135.72 | 8.0820 | 0.2410 | 101.75 |
| Dense w. DropBlock | 1.2468 | 5.3902 | 7.7579 | 7.6349 | 0.2119 | 135.72 | 8.0773 | 0.1934 | 101.75 |
| THOR ($k = N$) | 1.3110 | 4.8830 | 7.7620 | 7.6434 | 0.1439 | 135.73 | 8.0778 | 0.1607 | 101.76 |
| SMoE ($k = N$) | 1.1896 | 4.7982 | 7.7620 | 7.7537 | 0.1387 | 135.73 | 8.4291 | 0.1538 | 101.76 |
| SMoE-Dropout ($k = \frac{N}{2}$) | 1.1776 | 5.0220 | 5.6145 | 7.6372 | 0.1905 | 89.330 | 6.7693 | 0.1799 | 78.558 |
| SMoE-Dropout ($k = N$) | 1.1486 | 5.0220 | 7.7620 | 7.6293 | 0.1905 | 135.73 | 6.5491 | 0.1799 | 101.76 |

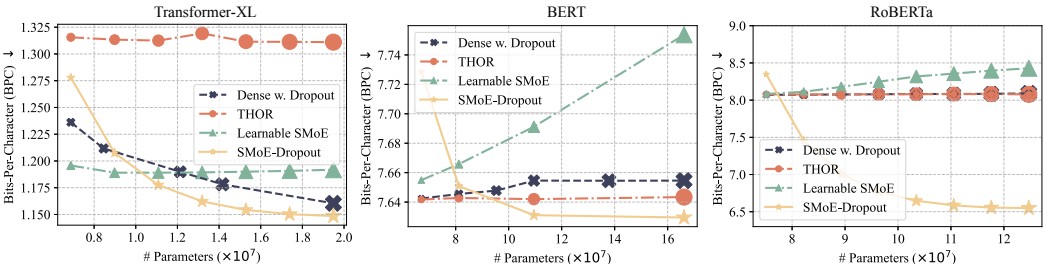

Figure 3: Testing performance over # parameter counts of {Transformer-XL, BERT, RoBERTa} networks on {`enwik8`, `BookCorpus`, `BookCorpus`} datasets, respectively. A smaller BPC suggests a better model.

collapse, bringing a *once-for-all* trade-off respected to inference resource availability. ❷ In contrast, learnable SMoE's and THOR's BPC are quickly saturated and deteriorated when adopting more experts, which implies the existence of expert redundancy (or representation collapse). The potential reasons for their substandard results are ($i$) the overfitting to fixed # experts utilized during training for learnable SMoE, ($ii$), and the consistency regularization between experts' predictions for THOR.

## 4.3 TRANSFER STUDY OF SMoE-DROPOUT: SELF-SLIMMABLE

We further investigate SMoE-Dropout and its intriguing "self-slimmable" property in a transfer learning scenario. Pre-trained models from Section 4.2 are *densely* fine-tuned on various downstream tasks, including text classification {`SST-2`} and challenging arithmetic & commonsense reasoning {`CSQA`, `ASDiv-A`, `MAWPS`, `SVAMP`}. The performance[2] is collected in Table 2. We find: equipped with SMoE-Dropout, Transformer-XL achieves $0.47\% \sim 2.43\%$ accuracy improvements on `SST-2`, BERT / RoBERTa obtain {$0.07\% \sim 9.72\%$, $0.42\% \sim 3.78\%$, $0.26\% \sim 1.30\%$, $1.09\% \sim 4.90\%$} and {$-$, $2.10\% \sim 5.88\%$, $0.07\% \sim 0.27\%$, $5.04\% \sim 5.93\%$} performance boosts on {`CSQA`, `ASDiv-A`, `MAWPS`, `SVAMP`} respectively, suggesting an enhanced transferability.

Similarly, we alter the model capacity during downstream fine-tuning. Starting from one pre-training, the SMoE-based method first calculates the selected times of each expert based on one feedforward pass with downstream data, then chooses the top activated experts to meet certain parameter budgets, and performs the subsequent dense fine-tuning. As displayed in Figure 4, our SMoE-Dropout has a continually increased accuracy or problem-solving rate when involving more parameters, and clearly surpasses the rest of approaches at parameter counts beyond 0.8, 8, and 10.5 ($\times 10^7$) for Transformer-XL, BERT, and RoBERTa respectively. It shows a flexible capacity adjustment, *i.e.*, "self-slimmable", according to the downstream resource constraint.

## 4.4 EXTRA INVESTIGATION AND ABLATION STUDY

***Q1:*** **When does SMoE-Dropout outperform other baselines?** *A1:* **Sufficient Model Capacity.**

To answer **Q1** and understand SMoE-Dropout's superiority in diverse scenarios, we investigate our proposal with different model capacities by varying model depth (*e.g.*, layers) & width (*e.g.*, experts).

---

[2]Due to limited computation resources, {our, official} pre-trained BERT/RoBERTa models are produced with {$10^5$, $10^6$} training iterations, {128, 256} batch size, {MLM, MLM and NSP} tasks, on {`BookCorpus` (800M words), `BookCorpus` (800M words) and `English Wikipedia` (2, 500M words)} dataset, respectively. The huge gap of pre-training outlays justifies the difference between our and official performance.

Table 2: Transfer performance {Accuracy (% ↑), Problem Solving Rate (% ↑)} of {Transformer-XL, BERT, RoBERTa} networks on {SST-2, CSQA, ASDiv-A, MAWPS, SVAMP} datasets. **All models are compared under the same number of parameter counts**. The same densely fine-tuning is adopted for all approaches, while THOR, SMoE, and SMoE-Dropout are tuned with half ($k = \frac{N}{2}$) or all ($k = N$) experts activated.

| Methods | Transformer-XL | BERT | | | | RoBERTa | | |
|---|---|---|---|---|---|---|---|---|
| | SST-2 | CSQA | ASDiv-A | MAWPS | SVAMP | ASDiv-A | MAWPS | SVAMP |
| Dense w. Dropout | 81.94 | 30.44 | 55.27 | 80.47 | 34.24 | 49.58 | 78.06 | 28.90 |
| THOR ($k = N$) | 81.13 | 20.79 | 52.52 | 79.95 | 30.43 | 53.36 | 77.86 | 28.44 |
| SMoE ($k = N$) | 79.98 | 29.27 | 55.88 | 80.73 | 33.15 | 52.10 | 77.86 | 27.98 |
| SMoE-Dropout ($k = \frac{N}{2}$) | 81.60 | 30.32 | 54.97 | 80.99 | 33.65 | 52.94 | 76.30 | 31.19 |
| SMoE-Dropout ($k = N$) | 82.41 | 30.51 | 56.30 | 81.25 | 35.33 | 55.46 | 78.13 | 33.94 |

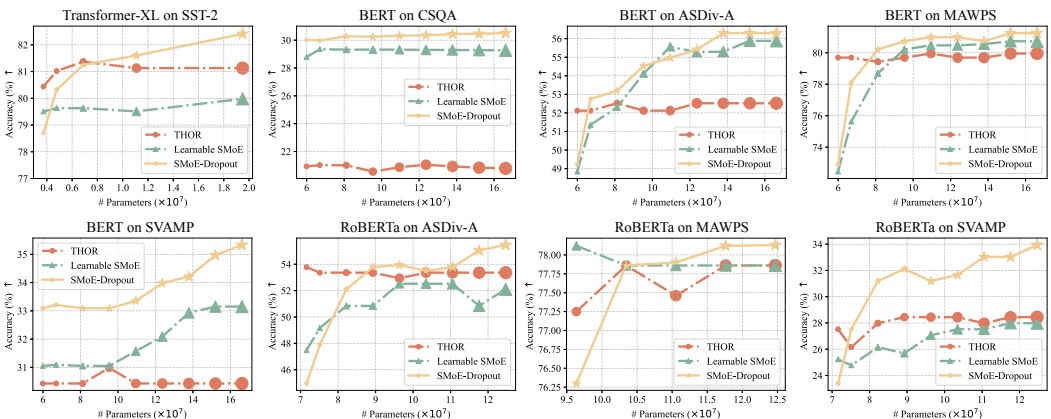

Figure 4: Transfer performance over # parameter counts of {Transformer-XL, BERT, RoBERTa} networks on downstream {SST-2, CSQA, ASDiv-A, MAWPS, SVAMP} datasets, respectively. Only the fine-tuning of Dense w. Dropout needs multiple pre-trained models with different amounts of network capacity.

**Model Depth - Different Number of Layers.** We conduct experiments on enwik8 dataset with Transformer-XL that has $2, 4, 8, 12$ layers and each layer is turned into the SMoE layer through a modularization. The comparison results of Densely Training w. Dropout and Training w. Learnable SMoE are reported in Figure 5 (a). We find that densely trained transformer performs the best when the network capacity is small like 2 layers, while with sufficiently large model capacity ($\geq 4$ layers), SMoE-Dropout demonstrates a consistent advantage compared to the others. Meantime, along with the increase of layers, the performance gap of SMoEs between the learned policy and our random policy keeps enlarging, signifying SMoE-Dropout's better scalability.

**Model Width - Different Number of Experts.** Similarly, we study the influence of model capacity by examining Transformer-XL with different widths of $2, 4, 8, 16$ experts. Results are summarized in Figure 5 (b). Consistent observations can be drawn that: ($i$) Densely Training w. Dropout outperforms SMoE-based training under small network widths such as $\leq 8$ experts; ($ii$) SMoE-Dropout presents enhanced performance when applied to large models with 16 experts; ($iii$) Learnable routing policies are effective with a small number of experts like $\leq 8$ experts, while it gets worse results than our random routing with a sufficient number of experts, *e.g.*, 16 experts.

*Q2: What is a better SMoE-Dropout design? A2:* **Random Weight Router; Later-layer SMoE.**

To answer **Q2**, we focus on the main constituents of SMoE-Dropout: *Modularization, Random Routing Policies*, and *Gradually Increased k*. Comprehensive ablations are depicted below.

**Ablation on Diverse Random Routing Policies.** An appropriate design of random routing policies determines the achievable performance of SMoE-Dropout. We compare our random initialized and fixed router to SMoE with fully random assignments (Zuo et al., 2022) and random hash SMoE with a pre-defined deterministic random assignment (Roller et al., 2021). Transformer-XL results on enwik8 are collected in Fig. 5 (c), where our proposed random routing obtains substantially lower BPC of $2.96 \sim 170.11 \ (\times 10^{-2})$ than the other two under different amounts of model parameters.

**Ablation on w./w.o. Gradually Increased $k$.** Figure 5 (d) investigates SMoE-Dropout variants with and without gradually increased $k$. We see that disabling the progressive manner of enlarg-

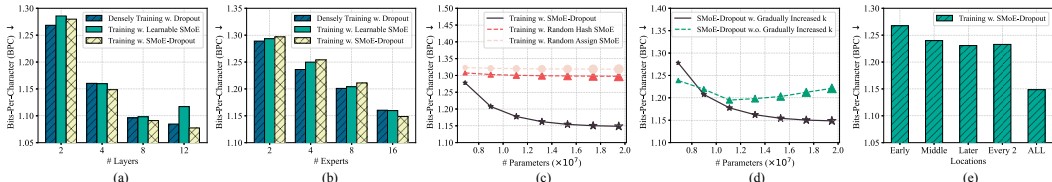

Figure 5: Extra studies about SMoE-Dropout. Testing BPC of Transformer-XL is collected on `enwik8`. (*a*) and (*b*) investigate diverse training mechanisms under different model depths and widths, respectively. (*c*) is the ablation of random routing policies. (*d*) examines the effects of gradually increased $k$. (*e*) studies the appropriate locations to insert SMoE expert layers.

ing the number of activated experts causes unsatisfied performance. Also, as a result, the "self-slimmable" property completely disappears, *e.g.*, adopting all model parameters leads to worse BPC.

**Ablation on Different Positions for Modularization.** It remains mysterious where is the best position to insert SMoE layers. To address this question, we perform modularization to different transformer layers and record their performance in Figure 5 (e). Specifically, given a 4-layer Transformer-XL, we compare four options: (*i*) *Early*, the first two layers are SMoE layers; (*ii*) *Middle*, the 2nd and 3rd layers are SMoE layers; (*iii*) *Later*, the last two layers are SMoE layers; (*iv*) *Every-2*, there is one SMoE layer every two transformer layers, *i.e.*, the 2nd and 4th layers. From the results, introducing SMoEs to later layers is in general more beneficial than modulizing earlier transformer layers. One possible reason is that shallow layers might capture common features that need to be shared across input samples. More dissections are left for future works.

*Q3:* **Extra benefits from SMoE-Dropout?** *A3:* **Improved Distillation and Less Overfitting.**

**Distilling into Single Expert on Downstream Tasks.** Besides all the benefits in pre-training inference and downstream transfer, we explore additional advantages of SMoE-Dropout under the distillation scheme that is usually preferred in resource-limited applications. As shown in Table 3, we distill all pre-trained Transformer-XLs into the same smaller variant with a single expert on the `SST-2` downstream task. Our algorithm produces the most distillable models among all four methods by a clear accuracy margin of $0.76\% \sim 1.89\%$.

**Overfitting.** We investigate the potential for overfitting to the training data distribution as model parameters increase in SMoE-Dropout, SMoE, and densely trained transformers. As shown in Figure 5 (a) and (b), experiments are conducted on `enwik8` with Transformer-XL, and three approaches are compared under the same parameter counts. We observe both SMoE-Dropout and Densely Training w. Dropout do not exhibit any indication of overfitting. That is, the performance is consistently improved as

Table 3: Distillation results of Transformer-XL on `SST-2`.

| Method | Accuracy (↑) |
|---|---|
| Dense w. Dropout | 81.25 |
| THOR | 80.76 |
| SMoE | 80.12 |
| SMoE-Dropout | 82.01 |

we increase the layers from 2 to 12 or experts from 2 to 16. In contrast, Training w. Learnable SMoE incurs BPC deterioration owing to overfitting when we expend the transformer to 12 layers, similar to the findings in Zoph et al. (2022). We attribute the reduced overfitting to the implicit regularization effect of SMoE-Dropout and Dropout.

## 5 CONCLUSION

In this paper, we present a novel plug-and-play SMoE-Dropout strategy for training over-parameterized transformers in full-capacity settings without collapse. We design a fixed and randomly initialized router to assign experts and gradually increase their number along with the training. As a result, our proposal provides an appealing "self-slimmable" property to large transformers during inference and downstream fine-tuning, depending on available resources. It implies alleviated representation collapse and delivers an in-situ trade-off between efficiency and performance. Extensive experiments across various combinations of network backbone and dataset, consistently demonstrate the significantly improved performance and training time savings from our algorithm. Future work includes the extension of other network architectures and tasks like vision recognition.

### ACKNOWLEDGEMENT

The research of ZW is in part supported by the US Army Research Office Young Investigator Award (W911NF2010240).

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

## A1 More Implementation Details

**Algorithm 1:** Concrete Dropout in a PyTorch-like style

```python
class ConcreteDropout(nn.Module):
    def __init__(self, weight_regularizer=1e-6,
                 dropout_regularizer=1e-5, init_min=0.1, init_max=0.1):
        super(ConcreteDropout, self).__init__()

        self.weight_regularizer = weight_regularizer
        self.dropout_regularizer = dropout_regularizer
        init_min = np.log(init_min) - np.log(1. - init_min)
        init_max = np.log(init_max) - np.log(1. - init_max)
        self.p_logit = nn.Parameter(torch.empty(1).uniform_(init_min,
            init_max))

    def forward(self, input, next_layer):
        p = torch.sigmoid(self.p_logit)
        # Apply Concrete Dropout
        output = self._concrete_dropout(input, p)
        # Feed forward through the next layer
        output = next_layer(output)
        # Calculate the weight regularizer
        sum_of_square = 0
        for param in next_layer.parameters():
            sum_of_square += torch.sum(torch.pow(param, 2))
        weights_regularizer = self.weight_regularizer * sum_of_square / (1
            - p)
        # Calculate the dropout regularizer
        dropout_regularizer = p * torch.log(p)
        dropout_regularizer += (1. - p) * torch.log(1. - p)
        input_dimensionality = input[0].numel()
        dropout_regularizer *= self.dropout_regularizer *
            input_dimensionality
        regularization = weights_regularizer + dropout_regularizer

        return output, regularization

    def _concrete_dropout(self, input, p):
        eps = 1e-7; temp = 0.1
        # Calculate the dropout probability matrix
        unif_noise = torch.rand_like(input)
        drop_prob = (torch.log(p + eps)
                - torch.log(1 - p + eps)
                + torch.log(unif_noise + eps)
                - torch.log(1 - unif_noise + eps))
        drop_prob = torch.sigmoid(drop_prob / temp)
        random_tensor = 1 - drop_prob
        retain_prob = 1 - p
        # Apply Concrete Dropout
        output = torch.mul(input, random_tensor)
        output /= retain_prob

        return output
```

**Algorithm 2:** DropBlock in a PyTorch-like style

```python
def drop_block(input, drop_prob, block_size):
    # Calculate the mask with zero value for block-wise elements
    mask = torch.rand_like(x).lt(drop_prob).float()
    mask = F.max_pool1d(mask, block_size, 1, block_size // 2)
    mask = 1 - mask
    output = input * mask # Apply dropout
    return output
```

**Details of Concrete Dropout and DropBlock.** In our experiments, we use Concrete-Dropout or DropBlock to replace the original dropout layer in MLP blocks. And for Concrete Dropout, we adopt the official implementation https://github.com/yaringal/ConcreteDropout/blob/master/concrete-dropout-pytorch.ipynb. For DropBlock, we follow the method in Ghiasi et al. (2018). And the PyTorch-style pseudo codes for both methods are presented in Algorithm 1 and 2.

## A2 MORE EXPERIMENT RESULTS

### A2.1 STABILITY ANALYSIS

To evaluate the stability of the improvement obtained by our SMoE-Dropout, we carry out further experiments of Transformer-XL on SST-2. The results are reported in Table A4, from which we can observe that our SMoE-Dropout achieves a statistically significant improvement of $0.93\% \sim 1.17\%$ accuracy gains compared with other SMoE-variants and the dense network, where there is no overlap between the error bars (one standard deviation).

Table A4: Test accuracy of Transformer-XL on SST-2. Both the average and standard deviation of accuracy are reported across 3 independent runs.

| Method | Accuracy ($\uparrow$) |
|---|---|
| Dense w. Dropout | $81.39 \pm 0.31$ |
| THOR ($k = $ N) | $81.15 \pm 0.55$ |
| SMoE ($k = $ N) | $81.20 \pm 0.50$ |
| SMoE-Dropout ($k = \frac{N}{2}$) | $82.03 \pm 0.26$ |
| SMoE-Dropout ($k = $ Ñ) | $82.32 \pm 0.14$ |

### A2.2 COMPARISON WITH LEARNABLE SMoEs W. GRADUALLY INCREASED $k$

Table A5 demonstrates that both random routing policy and progressively increasing the number of activated experts are beneficial for alleviating representation collapse and providing "self-slimmable" property, yet not as good as combining both. To be specific, when applying the strategy of progressively enlarging the number of activated experts, the learnable SMoEs suffer less representation collapse and achieve better performance, *i.e.*, $0.31\%$ higher accuracy. Meanwhile, We find that learnable SMoE with curriculum learning has the "self-slimmable" property only when activating experts from $k = 1$ to $k = 8$. However, the performance starts to degrade if using more experts like $k = 16$. As for our SMoE-Dropout with a random routing, it enjoys a better "self-slimmable" property from $k = 1$ to $k = 16$ (full model capacity), with up to $0.87\%$ higher accuracy on SST-2 across all scenarios, compared to its learnable variants.

Table A5: Testing accuracy (%) over # activated experts of Transformer-XL on SST-2.

| # Activated Experts | 1 | 2 | 4 | 8 | 16 |
|---|---|---|---|---|---|
| SMoE w.o. Gradually Increased $k$ | 80.06 | 80.79 | 80.58 | 80.99 | 81.20 |
| SMoE w. Gradually Increased $k$ | 79.40 | 81.02 | 81.13 | 81.71 | 81.51 |
| SMoE-Dropout | 79.02 | 81.25 | 82.00 | 82.03 | 82.32 |

### A2.3 TRANSFER STUDY ON MULTI-STEP REASONING TASKS

We conduct a further transfer study of the pre-trained BERT networks on a multi-hop question-answering dataset, HotpotQA Yang et al. (2018). And We use exact match (EM) accuracy to assess networks' performance. Following the same metric in Press et al. (2022), we calculate the compositionality gap, i.e., the gap of EM accuracy between multi-hop question answering and its all single-hop sub-questions Tang et al. (2020), of each network. As shown in Table A6, our SMoE-Dropout is beneficial for reducing the compositionality gap, which achieves the best performance with up to $0.30\%$ higher EM score and $0.30\%$ narrower compositionality gap, compared with the learnable SMoE and its dense counterpart.

Table A6: The EM score and compositionality gap of the pre-trained BERT networks on HotpotQA.

| Metrics | Dense w. Dropout | SMoE | SMoE-Dropout |
|---|---|---|---|
| EM Accuracy (%) | 15.10 | 14.90 | 15.20 |
| Compositionality Gap (%) | 14.90 | 15.00 | 14.70 |

