# OpenReview forum: "Sparse MoE as the New Dropout: Scaling Dense and Self-Slimmable Transformers"
_ICLR.cc/2023/Conference — ICLR 2023 notable top 25%_

### Official Review · Reviewer_DuzK · 2022-10-24

**Confidence:** 3
**Correctness:** 3
**Technical Novelty And Significance:** 3
**Empirical Novelty And Significance:** 3
**Recommendation:** 8

**Clarity, Quality, Novelty And Reproducibility:**

The writing clarity and technical quality are great. The method is simple but revisits MoE/dropout from a novel angle.

**Strength And Weaknesses:**

First of all, it is very interesting to see the connection drawn between the modern MoEs and the “old-fashioned” dropout. The former typically refers to both sparse training and sparse inference (at the same or similar sparsity), while the later conducts sparse training yet dense inference in full model capacity. Despites their formation similarity, traditionally they pursue different goals: this paper makes a revisit of them and neatly bridges their gap.

Their proposed solution, SMoE-Dropout, simply re-parameterizes a transformer with MoE layers for their MLPs, and then use a fixed random router to activate experts with a “curriculum learning” strategy, i.e., the number of activated experts gradually improved. Such training naturally yields a once-for-all in-situ trade-off between efficiency and performance when considering the deployment – a bonus that most current MoEs do not naturally enjoy. The method is also potentially generalizable to other non-transformer models.

The authors report very extensive empirical studies across various combinations of network architectures and datasets, as well as some transfer learning settings. They consistently demonstrate the significantly improved performance as well as training time savings from the SMoE-Dropout over a number of baselines. The experiment analyses are particularly well done and provide many useful insights on how SMoE-Dropout recipe works.

The following weakness points are identified, and clarifications are requested:
-	Is random routing really the major contributing factor in your recipe? What will happen if we apply the same curriculum learning of k to training learnable MoEs? Will they also become self-slimmable and suffer less collapse?
-	Another main critique I hold against this work is that in many experiments, the empirical gains of SMoE-Dropout seem just marginal over other baselines, although the performance increase trend seems consistent as more experts are activated.
-	 In Figure 1, it also appears that SMoE-Dropout performs worse than learnable MoE when the model is small.
-	I am also curious that, given the “self-slimming” property displayed on the same task, whether and how SMoE-Dropout could help address the compositional generalization challenge, that most MoEs nowadays can only solve with the assistance of clever prompting? For example, it would be very interesting to see the authors testing their trained models on multi-step commonsense or algorithm reasoning cases, perhaps with or without prompting.

**Summary Of The Paper:**

While many MoE research focused on improving routing policies to encourage expert diversity and specialization, this paper studies another aspect of expert scalability, i.e., gradually increasing the portion of experts being activated per time. The authors show that by making MoEs smoothly scalable to the number of experts, it can turn into a new structured dropout approach that effectively trains larger transformer models in their full capacity.

**Summary Of The Review:**

See the strength and weaknesses.

---

> ### Author Response · Authors · 2022-11-14
> **Responses to Reviewer DuzK [Cons 1-2]**
>
> Many thanks for rating our work as “novel” and “interesting”, acknowledging our methods “potentially generalizable”, and commenting our experiments as “extensive” and “provide many useful insights”. To address reviewer DuzK’s question, we provide point-wise responses below.
>
> **[Cons1: Apply Curriculum Learning to Learnable SMoE.]** We show that both random routing policy and curriculum learning method are beneficial for alleviating representation collapse and providing the self-slimmable property, yet not as good as combining both.
>
> To further convince reviewer DuzK, we carry out extra experiments of applying the same curriculum learning of $k$ to train learnable SMoEs with Transformer-XL on SST-2. The results are reported in Table R1, in which we can observe that equipped with curriculum learning, the learnable SMoEs suffer less representation collapse and achieve better performance, i.e., $0.31%$ higher accuracy with all experts activated on SST-2. We find that learnable SMoE with curriculum learning has the self-slimmable property only when activating experts from $k=1$ to $k=8$. However, the performance starts to degrade if using more experts like $k=16$.
>
> As for our SMoE-Dropout with a random routing, it enjoys a better “self-slimmable” property from $k=1$ to $k=16$ (full model capacity), with up to $0.87\%$ higher accuracy on SST-2 across all scenarios, compared to its learnble varints.
>
> Table R1: Testing accuracy over # activated experts of Transformer-XL on SST-2.
> |Experts|1|2|4|8|16|
> |:-:|:-:|:-:|:-:|:-:|:-:|
> |Learnable SMoE|$80.06$|$80.79$|$80.58$|$80.99$|$81.20$|
> |Learnable SMoE with curriculum learning|$79.40$|$81.02$|$81.13$|$81.71$|$81.51$|
> |SMoE-Dropout|$79.02$|$81.25$|$82.00$|$82.03$|$82.32$|
>
>
> **[Cons2: Marginal Improvements.]**  We politely point out that our improvements are consistent across various scenarios, and are statistically significant for parts of cases.
>
> - Extensive experiments with \{Transformer-XL, BERT, RoBERTa\} on multiple combinations of pre-training and downstream tasks, demonstrates **consistent** benefits from our proposed SMoE-Dropout, as shown in Table 1 and 2.
>
> - Additionally, we carry out extra experiments of Transformer-XL on the SST-2 task and report the average result and error bar (one standard deviation) of the performance by three different runs. As shown in Table R2, our method shows a statistically significant improvement of $0.93$\% $\sim$ $1.17$\% accuracy gains compared with other SMoE-variants and the dense network, where there is no overlap between the error bars.
>
> - Beyond performance gains, our approaches also enjoy computational savings. Take BERT on the pre-training task as an example. Our SMoE-Dropout obtains a moderate performance gain, and meanwhile, it shows up to $37$\% training time savings and $34$\% inference FLOPs reduction if only activating half experts at the evaluation stage.
>
> Table R2: Accuracy of Transformer-XL on the SST-2. All experiments are conducted by three different runs.
> |Task|Dense w. Dropout|THOR ($k=N$)|SMoE ($k=N$)|SMoE-Dropout ($k=\frac{N}{2}$)| SMoE-Dropout ($k=N$)|
> |:-:|:-:|:-:|:-:|:-:|:-:|
> |Accuracy (\%)|$81.39$ $\pm$ $0.31$|$81.15$ $\pm$ $0.55$|$81.20$ $\pm$ $0.50$|$82.03$ $\pm$ $0.26$|$82.32$ $\pm$ $0.14$|

---

> ### Author Response · Authors · 2022-11-14
> **Responses to Reviewer DuzK [Cons 3-4]**
>
> **[Cons3: Inferior Performance at Low Parameter Counts.]** Great catch. The reasons lie in several aspects:
>
> - The learnable SMoEs are known to suffer the routing collapse and load imbalance issue when containing a large number of experts, in which only a few experts will be frequently activated for most training samples [1]. Thus, when only activating this small part of experts (i.e., at low parameter counts), the learnable SMoEs are expected to achieve good performance. In the meantime, this is also why the learnable SMoEs do not have the self-slimmable property when using more experts, as other experts are not well-learned during training.
>
> - Moreover, take Transformer-XL on enwik8 as an example, although the learnable SMoE is better than other counterparts at low parameter counts (i.e., $0.7\sim1.0\times 10^7$ parameter counts), their performance is still in a bad range (i.e., BPC > $1.18$). Considering the BPC metric is the logarithm scale of original probability, there exists a substantial performance margin compared to activating 1/2 ($1.11\times 10^7$ parameter counts and $1.1776$ BPC), 2/3 ($1.53\times10^7$ parameter counts and $1.1542$ BPC), all ($1.95\times10^7$  parameter counts and $1.1486$ BPC) experts of networks empowered by our SMoE-Dropout. Note that, for inference with parameter counts larger than $1.02\times 10^7$ (with more experts activated), our models demonstrate clear performance advantages, thanks to the self-slimmable property.
>
>
>
> **[Cons4: Multi-Step Reasoning Tasks]** Thanks for the great suggestion. It’s really an interesting question and worth more further investigations. We conduct an extra transfer study of the pre-trained BERT networks on a multi-hop question-answering dataset, HotpotQA [2]. And We use exact match (EM) accuracy to assess networks’ performance. Following the same metric in [3], we calculate the compositionality gap, i.e., the gap of EM accuracy between multi-hop question answering and its all single-hop sub-questions [4], for each network. As shown in Table R3, our SMoE-Dropout is beneficial for reducing the compositionality gap, which achieves the best performance with up to $0.30$\% higher EM score and $0.30$\% narrower compositionality gap, compared with the learnble SMoE and its dense counterpart.
>
> Table R3: The EM score and compositionality gap of the pre-trained BERT networks on HotpotQA. A larger EM score and smaller compositionality gap indicate better performance.
> |Metrics|Dense w. Dropout|SMoE|SMoE-Dropout|
> |:-:|:-:|:-:|:-:|
> |EM|$15.10$|$14.90$|$15.20$|
> |Compositionality Gap|$14.90$|$15.00$|$14.70$|
>
> [1]  Outrageously Large Neural Networks: The Sparsely-Gated Mixture-of-Experts Layer.
>
> [2] HOTPOTQA: A Dataset for Diverse, Explainable Multi-hop Question Answering.
>
> [3] Measuring and Narrowing the Compositionality Gap in Language Models.
>
> [4] Do Multi-Hop Question Answering Systems Know How to Answer the Single-Hop Sub-Questions?

---

### Official Review · Reviewer_qsko · 2022-10-24

**Confidence:** 4
**Correctness:** 3
**Technical Novelty And Significance:** 2
**Empirical Novelty And Significance:** 2
**Recommendation:** 8

**Clarity, Quality, Novelty And Reproducibility:**

Paper would benefit from explaining the Dropout integration along with the expert activation schedule a bit more. It is unclear why the linear activation schedule would be the optimal in this setup.

Novelty of this work is somewhat limited given that others have tried out similar notion of gradual expert activation (see https://arxiv.org/abs/2112.14397v1).


Nit-picks
- Page 2, Para 3, "... SMoE-Dropout ican be ... "
- Page 3, Para 4, "... inferences. Meantime, the presented .."
- Page 4, Eq.1, 'TopK( softmax(G(x), k) )' should change to 'TopK( softmax(G(x)), k )'
- Term "progressive" might suit better in this context rather than self-slimmable






**Strength And Weaknesses:**


Strengths:
- "Self-Slimmable" -- the consistent boost in transformer performance with an increase in activated experts during inference and downstream fine-tuning
- Extensive empirical evaluation that demonstrates the improvements on pre-training and reasoning tasks

Weaknesses:
- Does not explain the proposal in details (see below)
- Limited novelty ( progressive activation of experts by https://arxiv.org/abs/2112.14397v1, other works have looked at progressive activation of kernels in CNNs such as Once-for-All networks  )


Questions for Authors:

(1) SMoE-Dropout needs to be explained in more detail than what is presented in 3.2. What does it mean to say that the router network is
 - randomly initialized
 - and fixed
 - and implicitly optimized during training

 Either it is fixed or you train the network. May be elaborate any points that I have missed.

(2) How do you integrate the Concrete-Dropout and DropBlock in the proposed SMoE-Dropout? A simple pseudo-code that shows these steps end-to-end would help a lot to clarify things.

(3) Why is only the linear schedule used to gradually increase during the training? How about other ways to progressively activate the experts? How does dropout and expert activation schedule interplay?

(4) How is Figure~1 generated?
   - For SMoE-Dropout --> for the same SMoE, you increase the number of activate experts --> yielding in higher parameter count.
   - How are green and black curves generated?
   - Since black represents the dense network --> it should be a collection of different dense models
   - Why is the proposed scheme worse in the low parameter count regime?


**Summary Of The Paper:**

Sparse Mixture-of-Experts (SMoE) have been widely used to increase the representation capacity of transformers with similar computational cost as dense transformers. SMoE consists of many experts, only few of which are activate for any given input example. This increased representational power comes with two issues:
(a) Representation Collapse -- as the latent space representations get centered around experts leading to lots of redundancy and sub-optimal performance
(b) Poor Scalability -- During inference and downstream fine-tuning, increasing the number of active experts does not increase the performance due to overfitting to the active experts during training

This work addresses these issues by introducing SMoE-dropout. It consists of randomly initialized and fixed router network that activates the experts. It gradually increases the number of experts during the training. This yields SMoE that have "self-slimmable" property, i.e., offering consistent boosted performance for transformers with an increase in activated experts during inference and downstream fine-tuning. Empirical evaluations show that SMoE transformers trained with SMoE-dropout yields consistent improvements over their dense counterparts on various reasoning tasks.

**Summary Of The Review:**

Overall the self-slimmable property is indeed important for a sparse mixture of experts model and the proposed architecture helps in achieving the same. The paper is its current state needs to add more details to the method and explain some of the design choices. It is also lacking in novelty as mentioned earlier.


----


I've read other reviewer comments and author's response. I'll increase my score accordingly.

---

> ### Author Response · Authors · 2022-11-14
> **Responses to Reviewer qsko [Cons 1-2]**
>
> Thank reviewer qsko for the valuable comments. We elaborate more details of our method as well as explanations of the design choices. Meanwhile, point-to-point responses are provided as below.
>
> **[Cons1: More Explanation of the Proposed Method.]** To clarify reviewer qsko’s confusion, we introduce some simple notations. Specifically, let $\theta^r=\theta^r_0$ and $\theta=\theta_0$ denote the model parameters of the router network and other network components (e.g., experts), respectively. They are randomly initialized as $\theta^r_0$ and $\theta_0$. Throughout the training process, we only update $\theta$ while keeping $\theta^r=\theta^r_0$ unchanged. This is also why we describe the router as “randomly initialized and fixed”.
>
> However, with the update of $\theta$, the produced intermediate token embeddings are changed in each iteration. Note that the routing assignment is dependent on both $\theta^r_0$ (router weights) and the intermediate token embeddings (router inputs), which is updated accordingly due to the optimzied embeddings. Therefore, we say our routing policy is “implicitly optimized” rather than fully random.
>
> **[Cons2: Implementation of Concrete-Dropout and DropBlock.]** Thanks for the suggestion. In our experiments, we use Concrete-Dropout or DropBlock to replace the original dropout layer in MLP blocks. And for Concrete Dropout, we adopt the official implementation (https://github.com/yaringal/ConcreteDropout/blob/master/concrete-dropout-pytorch.ipynb). For DropBlock, we follow the method in [1]. And the PyTorch-style pseudo codes for both methods are shown as the following:
>
>
> ```
> class ConcreteDropout(nn.Module):
>     def __init__(self, weight_regularizer=1e-6,
>                  dropout_regularizer=1e-5, init_min=0.1, init_max=0.1):
>         super(ConcreteDropout, self).__init__()
>
>         self.weight_regularizer = weight_regularizer
>         self.dropout_regularizer = dropout_regularizer
>         init_min = np.log(init_min) - np.log(1. - init_min)
>         init_max = np.log(init_max) - np.log(1. - init_max)
>         self.p_logit = nn.Parameter(torch.empty(1).uniform_(init_min, init_max))
>
>     def forward(self, input, next_layer):
>         p = torch.sigmoid(self.p_logit)
>         # Apply Concrete Dropout
>         output = self._concrete_dropout(input, p)
>         # Feed forward through the next layer
>         output = next_layer(output)
>         # Calculate the weight regularizer
>         sum_of_square = 0
>         for param in next_layer.parameters():
>             sum_of_square += torch.sum(torch.pow(param, 2))
>         weights_regularizer = self.weight_regularizer * sum_of_square / (1 - p)
>         # Calculate the dropout regularizer
>         dropout_regularizer = p * torch.log(p)
>         dropout_regularizer += (1. - p) * torch.log(1. - p)
>         input_dimensionality = input[0].numel()
>         dropout_regularizer *= self.dropout_regularizer * input_dimensionality
>         regularization = weights_regularizer + dropout_regularizer
>
>         return output, regularization
>
>     def _concrete_dropout(self, input, p):
>         eps = 1e-7; temp = 0.1
>         # Calculate the dropout probability matrix
>         unif_noise = torch.rand_like(input)
>         drop_prob = (torch.log(p + eps)
>                 - torch.log(1 - p + eps)
>                 + torch.log(unif_noise + eps)
>                 - torch.log(1 - unif_noise + eps))
>         drop_prob = torch.sigmoid(drop_prob / temp)
>         random_tensor = 1 - drop_prob
>         retain_prob = 1 - p
>         # Apply Concrete Dropout
>         output  = torch.mul(input, random_tensor)
>         output /= retain_prob
>
>         return output
> ```
>
>
> ```
> def drop_block(input, drop_prob, block_size):
>     # Calculate the mask with zero value for block-wise elements
>     mask = torch.rand_like(x).lt(drop_prob).float()
>     mask = F.max_pool1d(mask, block_size, 1, block_size // 2)
>     mask = 1 - mask
>     output = input * mask # Apply dropout
>     return output
> ```
>
>
> [1] DropBlock: A regularization method for convolutional networks.

---

> ### Author Response · Authors · 2022-11-14
> **Responses to Reviewer qsko [Cons 3-6]**
>
>
> **[Cons3: Other Ways to Progressively Activate the Experts.]** We conduct extra comparison experiments with two other schedules, exponential increase, i.e., $N_t = round\Big((N_s - N_e) \cdot \gamma^{x(t)} + N_e\Big)$, and cosine increase, i.e., $N_t = round\Big((N_s - N_e) \cdot cos(\frac{\pi \cdot t}{2 \cdot t_{total}}) + N_e\Big)$, where $N_s$, $N_e$, and $N_t$ represent the initial, final, and current number of activated experts, respectively. And $t$, $t_{total}$ is the current and total optimization steps. $\gamma$ is the hyperparameter, which equals $0.99$. And $x(t) = round(\frac{t}{t_{interval}})$ with $t_{interval}=1300$ in our implementation. The results are reported in Table R1. Despite the simplicity, the linear schedule shows superior performance.
>
> Table R1: Test performance of different schedules in SMoE-Dropout. We assess the performance via the Bits-Per-Character (BPC) metric, where a smaller BPC value indicates a better performance.
> |Schedule|SMoE-Dropout ($k=\frac{N}{2}$)| SMoE-Dropout ($k=N$)|
> |:-:|:-:|:-:|
> |Linear|$1.1776$|$1.1486$|
> |Exponential|$1.2167$|$1.1877$ |
> |Cosine|$1.2142$|$1.1858$|
>
> **[Cons 4: How Does Dropout and Expert Activation Schedule Interplay?]** In our paper, we do not combine Droput and SMoE. Based on the finding in [2], directly integrating Dropout and SMoE results in inferior performance. The reason that we name our method as SMoE-Dropout is they share similar design philosophy - “randomly” activating part of the model and disabling the rest part.
>
> **[Cons 5: Explanation of Figure 1.]** Yes, for SMoE-Droput, we use the same SMoE and increase the number of activated experts for inference, yielding higher parameter counts.
>
> Similarly, the green curve is generated from the same learned SMoE model. And we evaluate the test performance with different numbers of experts activated (thereby different parameter counts).
>
> Yes, the black curve is generated from a collection of different dense models. They have different width and dimension sizes of the intermediate features in the MLP blocks.
>
> **[Cons 6: Inferior Performance at Low Parameter Counts.]** Great catch. The reasons lie in several aspects:
>
> - The learnable SMoEs are known to suffer the routing collapse and load imbalance issue when containing a large number of experts, in which only a few experts will be frequently activated for most training samples [3]. Thus, when only activating this small part of experts (i.e., at low parameter counts), the learnable SMoEs are expected to achieve good performance. In the meantime, this is also why the learnable SMoEs do not have the self-slimmable property when using more experts, as other experts are not well-learned during training.
>
> - Moreover, take Transformer-XL on enwik8 as an example, although the learnable SMoE is better than other counterparts at low parameter counts (i.e., $0.7\sim1.0\times 10^7$ parameter counts), their performance is still in a bad range (i.e., BPC > $1.18$). Considering the BPC metric is the logarithm scale of original probability, there exists a substantial performance margin compared to activating 1/2 ($1.11\times 10^7$ parameter counts and $1.1776$ BPC), 2/3 ($1.53\times10^7$ parameter counts and $1.1542$ BPC), all ($1.95\times10^7$  parameter counts and $1.1486$ BPC) experts of networks empowered by our SMoE-Dropout. Note that, for inference with parameter counts larger than $1.02\times 10^7$ (with more experts activated), our models demonstrate clear performance advantages, thanks to the self-slimmable property.
>
> [2] Designing Effective Sparse Expert Models.
>
> [3] Outrageously Large Neural Networks: The Sparsely-Gated Mixture-of-Experts Layer.

---

> ### Author Response · Authors · 2022-11-14
> **Responses to Reviewer qsko [Cons 7-8]**
>
> **[Cons 7: Somewhat Limited Novelty.]** We respectfully disagree and we see our work novel. Reasons lie in below:
>
> - Firstly, [4] targets the training difficulty of learnable SMoEs. We aim to a totally different problem of how to train big dense transformer-based networks.
>
> - In our paper, we propose a new approach by first performing modularization to the big dense model and then optimizing it by leveraging random SMoEs as an auxiliary tool.
>
> - Additionally, [4] focus on learnable SMoEs. And learnable SMoEs normally have concerns about load balance and representation collapse. With a different target, we study random routing strategies, which naturally mitigate these issues to some extent.
>
> - Beyond that, our SMoE-Dropout provides a “self-slimmable” property for the first time, which offers a consistently boosted inference performance for models with more experts activated. Such “slimmable” property enables scaling transformers to the full capacity without collapse, bringing a once-for-all trade-off respected to inference resource availability.
>
> - As nicely summarized by reviewer 3ZBm, our proposal is “a new way to train big dense models by leveraging MoEs as an auxiliary tool”. Our work is also acknowledged by reviewer EKmy, as “ a novel plug-and-play strategy”; and by reviewer DuzK as “a novel angle”.
>
> **[Cons 8: Typos.]** Thanks for the detailed writing suggestions. We have addressed all inappropriate expressions and typos in the updated version.
>
>
> [4] Dense-to-Sparse Gate for Mixture-of-Experts.

---

> ### Author Response · Authors · 2022-11-21
> **Sincerely expecting further discussions from Reviewer qsko**
>
> Dear Reviewer **qsko**,
>
> We thank reviewer **qsko** for the time of reviewing and the constructive comments. We really hope to have a further discussion with reviewer **qsko** to see if our response solves the concerns.
>
> In our response, we have (1) provided more details about our methods, baseline Dropout, Figure 1; (2) conducted additional experiments about the expert activation schedule; (3) clarified all other confusions.
>
> We genuinely hope reviewer **qsko** could kindly check our response. Thank you!
>
> Best wishes,
>
> Authors

---

> ### Author Response · Authors · 2022-11-25
> **Sincerely expecting further discussions from Reviewer qsko**
>
> Dear Reviewer **qsko**,
>
> We genuinely thank reviewer **qsko** for your time. We hope our previous responses have addressed your concerns.
>
> As the discussion period is approaching its end, we would really appreciate it if you could kindly let us know whether there are any further questions. We will be more than happy to address them fully.
>
> Your Sincerely,
>
> Authors

---

> > ### Comment · Reviewer_qsko · 2022-11-26
> > **Thank you for the detailed response.**
> >
> > Thank you for the detailed response. It has helped clear some of my confusion around the proposed mechanism. I'll increase my score. I would suggest the authors to incorporate some of the response in the comments (in my review as well as other reviewers) in the paper to strengthen the work.

---

> > > ### Author Response · Authors · 2022-11-26
> > > **Thank you for increasing the score**
> > >
> > > Dear Reviewer **qsko**,
> > >
> > > Many thanks for all the helpful comments and positive re-assessment. We really appreciate reviewer **qsko** for increasing our score.
> > >
> > > Yes. Most of the responses have already been included in our current revision like more methodology details, additional experiments, and their analyses. In our final version, we will further refine and improve our paper based on all reviewer comments.
> > >
> > > We are again very thankful for your time and support!
> > >
> > > Best wishes,
> > >
> > > Authors

---

### Official Review · Reviewer_3ZBm · 2022-10-25

**Confidence:** 4
**Correctness:** 3
**Technical Novelty And Significance:** 3
**Empirical Novelty And Significance:** 3
**Recommendation:** 8

**Clarity, Quality, Novelty And Reproducibility:**

It seems the proposed method is not a new training recipe for MoEs, but rather a new way to train big dense models by leveraging MoEs as an auxiliary tool. This confused me initially. Clarifying this point better in abstract would help readers set their expectation.

**Strength And Weaknesses:**

Main Strength Points
(1)	SMoE-Dropout facilitates randomly and sparsely activated structure of network modules, playing an implicit regularization role similar to dropout – but more scalable to large transformer models. It has very low overhead and is very easy to use.
(2)	SMoE-Dropout naturally provides a “self-slimmable” property offering consistent performance boost for transformers with an increase in activated experts during inference and downstream fine-tuning. The transfer gains are especially obvious.
(3)	Experiments are very extensive and thorough, including a variety of tasks, architectures, datasets, and strong baselines beyond vanilla dropout. Both standard training and transfer learning are reported, accompanied with in-depth ablation studies. The experiment and analysis are very well organized and read convincing.

Weakness and/or clarification:
(1)	Does SMoE-training cost higher memory overhead, compared to dense training?
(2)	It is very interesting to see that the SMoE-trained model can be distilled into a single-expert backbone. But details are lacked on how the distillation was done: is it common logit matching or something else? Also, how other MoE-variants will performance under distillation?
(3)	For learnable MoE overfitting and collapse, it is an already well-known phenomenon that can also be mitigated by various diversity-regularized routers. It would be nice to compare SMoE-dropout with some of them.


**Summary Of The Paper:**

SMoE-Dropout is motivated by the representation collapse and expert scalability issues observed in learned MoEs, by dynamically adapting number of activated experts as needed. It consists of a randomly initialized and fixed router network to activate experts and gradually increase their number as training progresses over time.

**Summary Of The Review:**

The paper proposed a new method for dynamically adapting number of activated experts as needed and the overall quality is good. If the  questions as described above are well addressed, I tend to increase my score.

---

> ### Author Response · Authors · 2022-11-14
> **Responses to Reviewer 3ZBm**
>
> Many thanks to Reviewer 3ZBm for acknowledging that our method is “very low overhead and very easy to use”, our transfer gains are “especially obvious”, our experiments are “very extensive and thorough”, and our analyses are “very well organized and convincing”. To address reviewer 3ZBm’s questions, we provide point-wise responses below.
>
>
> **[Cons1: Does SMoE-training Cost Higher Memory Overhead?]** No. The memory overhead of SMoE-training even with all experts activated is nearly the SAME as the one of directly dense training. The key reason is that our methods divide the large dense into smaller experts via modulization, rather than introducing extra experts. To be specific:
>
> - As shown in Figure 2 and Section 3.2, the modulization step in our SMoE-Dropout turns a large densely connected MLP into multiple smaller MLPs (i.e., experts) with the same size. Thus the overall parameter counts and intermediate embedding size of SMoE-Dropout  (when all experts are activated) is the SAME as its dense counterpart, leading to the same memory overhead. Although the routing network may cost extra memory outlay, the parameter count of the router is far from that of the main network, e.g., for transformer-xl, the router only contains $0.84\%$ of the whole parameters, which is negligible.
>
> - Beyond that, in our SMoE-Dropout, we progressively increase the number of activated experts and only part of the experts will be activated during most of the training period, in which we require less memory overhead than directly training the large dense model.
>
> **[Cons2: Distillation Details and Results of Other MoE-variants.]** Thanks for the question. We follow the original knowledge distillation technique [1] and distill the pre-trained networks into the same smaller variant with a single expert via logit matching. The results are reported in Table 3. We can observe that our SMoE-Dropout achieves an extra accuracy gain (up to $1.89\%$) compared with other SMoE-variants (i.e., THOR, SMoE) as well as the dense counterpart.
>
> **[Cons3: Comparison of Other Diversity-Regularized Routers]** Thanks for the constructive suggestion. We conduct an extra experiment for the comparison.
>
> - We adopt the regularization technique in [2] for the learnable SMoEs of Transformer-XL on SST-2, which obtains a $0.62\%$ improvement of accuracy with all experts selected but remains $0.50\%$ gap from SMoE-Dropout. Beyond that, for inference with parameter counts larger than $4.8 \times 10^6$ (more experts activated), our SMoE-Dropout demonstrates a consistent improvement (up to $0.50\%$) compared to the learnable SMoE with regularization, whose performance is quickly saturated when adopting more experts. Thus our SMoE-Dropout enjoys a better self-slimmable property, which brings a once-for-all trade-off respected to inference resource availability. Meantime, we are willing to investigate more diversity-regularized approaches in our future version, if reviewer 3ZBm could kindly point out some.
>
> - We see that most previous regularization-based works enhance the diversity of representations [2,3] by leveraging auxiliary losses, while our SMoE-Dropout investigates the problem through a different perspective by progressively updating network architectures during training. They are orthogonal to each other and can neatly be combined. We will explore this interesting point in our future work.
>
> **[Cons4: Clarifying of the Proposed Method in Abstract]** Thanks. We’ve revised the abstract section in the updated draft, in which we clarify the main focus of our SMoE-Dropout is to provide a new approach for training large-scale dense networks, rather than a better training recipe for SMoEs. Modifications are marked in blue, i.e., “our work focuses on exploring the overlooked scalability bottleneck of SMoEs, and leveraging it as an auxiliary tool to effectively benefit scaling large-scale dense transformers.”
>
> [1] Distilling the Knowledge in a Neural Network.
>
> [2] Switch Transformers: Scaling to Trillion Parameter Models with Simple and Efficient Sparsity.
>
> [3] ​​Gshard: Scaling giant models with conditional computation and automatic sharding.

---

> > ### Comment · Reviewer_3ZBm · 2022-11-21
> > **Thanks**
> >
> > Thanks. The responses have well addressed my concerns. The proposed SMoE-Dropout solves the representation collapse and expert scalability issues observed in learned MoEs. The memory overhead of SMoE-training even with all experts activated is nearly the SAME as the one of directly dense training. Thus, I increase my score.

---

> > > ### Author Response · Authors · 2022-11-26
> > > **Thank you for increasing the score**
> > >
> > > Dear Reviewer **3ZBm**,
> > >
> > > Many thanks for all the constructive comments and very positive evaluations. We really appreciate reviewer **3ZBm** for increasing our score.
> > >
> > > We are again very thankful for your time and support!
> > >
> > > Best wishes,
> > >
> > > Authors

---

### Official Review · Reviewer_EKmy · 2022-10-25

**Confidence:** 3
**Correctness:** 3
**Technical Novelty And Significance:** 3
**Empirical Novelty And Significance:** 3
**Recommendation:** 8

**Clarity, Quality, Novelty And Reproducibility:**

All look good to me. Authors promised to release models and codes: it would help if the authors could respond to be clearer what they plan to release: all experiments versus some, pre-trained model versus training script, etc.

**Strength And Weaknesses:**

Pros
-	SMoE-Dropout demonstrates an attractive “self-slimming” property during inference and downstream fine-tuning, which delivers a once-for-all in-situ trade-off between efficiency and performance
-	Like classical dropout, SMoE-Dropout is able to mitigate the representation collapse that standard MoEs usually suffer from, i.e., activating more experts do not improve or even hurt performance.
-	Applying SMoE-Dropout is extremely cheap as only random router is adopted
-	Transfer study is strong, which provides another evidence that more pre-training information is effectively retained by the full transformer capacity. It is further validated in ablation studies 4.4

Cons:
-	The performance gain of SMoE-Dropout is sometimes marginal, such as on BeRT (section 4.1). Moreover, as seen from Figure 1 and Figure 3, it seems SMoE-Dropout does sacrifice performance at low parameter counts compared to Learnable MoE, why?
-	When breaking a single-stream models into MLP MoEs (the modularization step in section 3.2), how to decide the number of MLP experts needed, i.e., N? This seems to be an important hyperparameter but not discussed. Also, why not applying SMoE-Dropout to MLP layers but not self-attention?
-	For training with Learnable SMoE, it is further unclear how the authors selected its k?
-	Why training time is the same between SMoE-dropout k=N/2 and k=N, in Table 1?
-	Table 1 again: are all those numbers averaged across three datasets? The authors didn’t explain


**Summary Of The Paper:**

This paper presents a novel plug-and-play strategy for training large transformer models, which leverages sparse MoEs in a dropout-like manner to scale transformers to better performance in their full capacity without collapse. The method is simple, and the experiments are thorough.

**Summary Of The Review:**

Overall, I would tend to accept this paper. If the author can address my concerns. I would be more convinced.

---

> ### Author Response · Authors · 2022-11-14
> **Responses to Reviewer EKmy [Cons 1-3]**
>
> We’re glad that reviewer EKmy has a positive initial impression of our work, and acknowledges our “transfer study is strong”. Many thanks for the constructive feedback especially for the questions about our detailed settings, which helps us to further enrich and improve our paper. To address reviewer EKmy’s questions, we provide pointwise responses below.
>
>
> **[Cons 1: Marginal Improvements.]**  We politely point out that our improvements are consistent across various scenarios, and are statistically significant for parts of cases.
>
> - Extensive experiments with \{Transformer-XL, BERT, RoBERTa\} on multiple combinations of pre-training and downstream tasks, demonstrates **consistent** benefits from our proposed SMoE-Dropout, as shown in Table 1 and 2.  Moreover, as one of the key advantages, SMoE-Dropout grants the trained transformer with a “self-slimmable” property --- “the consistent boost in transformer performance with an increase in activated experts during inference and downstream fine-tuning” as nicely summarized by reivewer qsko.
>
> - Additionally, we carry out extra experiments of Transformer-XL on the SST-2 task and report the average result and error bar (one standard deviation) of the performance by three different runs. As shown in Table R1, our method shows a statistically significant improvement of $0.93$\% $\sim$ $1.17$\% accuracy gains compared with other SMoE-variants and the dense network, where there is no overlap between the error bars.
>
> - Beyond performance gains, our approaches also enjoy computational savings. Take BERT on the pre-training task as an example. Our SMoE-Dropout obtains a moderate performance gain, and meanwhile, it shows up to $37$\% training time savings and $34$\% inference FLOPs reduction if only activating half experts at the evaluation stage.
>
>
> Table R1: Accuracy of Transformer-XL on the SST-2. All experiments are conducted by three different runs.
> |Task|Dense w. Dropout|THOR ($k=N$)|SMoE ($k=N$)|SMoE-Dropout ($k=\frac{N}{2}$)| SMoE-Dropout ($k=N$)|
> |:-:|:-:|:-:|:-:|:-:|:-:|
> |Accuracy (\%)| $81.39$ $\pm$ $0.31$ | $81.15$ $\pm$ $0.55$ | $81.20$ $\pm$ $0.50$ | $82.03$ $\pm$ $0.26$ | $82.32$ $\pm$ $0.14$ |
>
>
>
> **[Cons 2: Inferior Performance at Low Parameter Counts.]** Great catch. The reasons lie in several aspects:
>
> - The learnable SMoEs are known to suffer the routing collapse and load imbalance issue when containing a large number of experts, in which only a few experts will be frequently activated for most training samples [1]. Thus, when only activating this small part of experts (i.e., at low parameter counts), the learnable SMoEs are expected to achieve good performance. In the meantime, this is also why the learnable SMoEs do not have the self-slimmable property when using more experts, as other experts are not well-learned during training.
>
> - Moreover, take Transformer-XL on enwik8 as an example, although the learnable SMoE is better than other counterparts at low parameter counts (i.e., $0.7\sim1.0\times 10^7$ parameter counts), their performance is still in a bad range (i.e., BPC > $1.18$). Considering the BPC metric is the logarithm scale of original probability, there exists a substantial performance margin compared to activating 1/2 ($1.11\times 10^7$ parameter counts and $1.1776$ BPC), 2/3 ($1.53\times10^7$ parameter counts and $1.1542$ BPC), all ($1.95\times10^7$  parameter counts and $1.1486$ BPC) experts of networks empowered by our SMoE-Dropout. Note that, for inference with parameter counts larger than $1.02\times 10^7$ (with more experts activated), our models demonstrate clear performance advantages, thanks to the self-slimmable property.
>
> **[Cons 3: How to Choose the Number of Experts?]** Thanks for the question. We conduct a further ablation study about the number of MLP experts needed. Given the same dense network, we turn its densely connected MLP into $N$ smaller MLPs with the same size, in which $N=2,4,8,16,32$, respectively. Table R2 shows the results of Transformer-XL on enwik8.  In general, more experts produce a better result when $N\le16$. And it seems that $N=16$ is a “sweet point” for achievable performance.
>
> In addition, a larger number of experts (i.e., dividing the large dense into more pieces) provides a more **flexible** and **fine-grained** in-situ trade-off between inference outlay and model performance based on our self-slimmable property.
>
> Table R2: Test performance of Transformer-XL on enwik8 with different numbers of experts. We assess the performance via the Bits-Per-Character (BPC) metric, where a smaller BPC value indicates a better performance.
> |Experts|2|4|8|16|32|
> |:-:|:-:|:-:|:-:|:-:|:-:|
> |BPC|$1.1942$|$1.1933$|$1.1891$|$1.1486$|$1.1716$|
>
>
> [1] Outrageously Large Neural Networks: The Sparsely-Gated Mixture-of-Experts Layer.

---

> > ### Comment · Reviewer_EKmy · 2022-11-18
> > **Thanks for the detailed rebuttal**
> >
> > My problems have been well addressed. Thus, I increase my score.

---

> > > ### Author Response · Authors · 2022-11-26
> > > **Thank you for increasing the score**
> > >
> > > Dear Reviewer **EKmy**,
> > >
> > > We sincerely appreciate all the helpful feedback and very positive evaluations from reviewer **EKmy**.
> > >
> > > We are again very thankful for your time & support, and for increasing our score!
> > >
> > > Best wishes,
> > >
> > > Authors

---

> ### Author Response · Authors · 2022-11-14
> **Responses to Reviewer EKmy [Cons 4-8]**
>
> **[Cons 4: Why Only Applying SMoE-Dropout to MLP Layers?]** Thanks for the question.  As mentioned in section 3.1, most existing SMoE works deal with the MLP component in transformers because MLPs constitute roughly 2/3 of total model parameter counts storing significant amounts of learned knowledge as memory networks [2,3]. And to our best knowledge, only one previous work applies such spirit to attention modules for a specific purpose, i.e., handling the task of multiple modalities and fusing their knowledge [4].
>
> To further address reviewer EKmy’s question, we apply SMoE-Dropout to both the MLP and attention blocks in Transformer-XL and train it on enwik8, which obtains $1.1712$ BPC on test-set, slightly worse than directly dense training. The result seems to indicate that SMoE-type attention block has limited contributions in our single modality learning scenarios.
>
> **[Cons 5: How to Select K for Learnable SMoEs?]** For a fair comparison, we choose $k=12$ for the learnable SMoE in our experiments, which has the same training computation costs as our SMoE-Dropout. It is also worth mentioning that training learnable SMoEs with $k=12$ ($1.1896$ BPC) performs better than its conventional variant with $k=2$ ($1.1920$ BPC).
>
> **[Cons 6: Same Training Time of SMoE-Dropout with Different Numbers of Activated Experts]** Because SMoE-Dropout ($k=\frac{N}{2}$) and SMoE-Dropout ($k=N$) is the *SAME* trained network but evaluated with different numbers of activated experts. Thus the training time is the same.
>
> By providing inference results with half and all experts activated, the goal is to show (i) our self-slimmable property that using more experts leads to consistently improved performance; (ii) and an in-situ trade-off between inference outlay and model performance.
>
> **[Cons 7: Are Numbers in Table 1 Averaged across Three Datasets?]** No, they are separate numbers on each dataset. As stated in the caption of Table 1, we conduct experiments of three architectures on {enwik8, BookCorpus, BookCorpus}, **respectively**. And each number represents the corresponding result of one architecture on one dataset.
>
>
> **[Cons 8: Model and Code Release]** Thanks for the great suggestion. We’ve updated the code in the supplementary, which contains the training scripts of experiments in Table 1,2, and Figure 5. And we will release the corresponding pre-trained models after acceptance.
>
> [2] Transformer feed-forward layers are key-value memories.
>
> [3] Knowledge neurons in pretrained transformers.
>
> [4] One Model, Multiple Modalities: A Sparsely Activated Approach for Text, Sound, Image, Video and Code.

---

### Author Response · Authors · 2022-11-14
**General Responses**

We sincerely appreciate all reviewers’ time and efforts in reviewing our paper. And we also thank all reviewers for the constructive comments and suggestions, which helped further improve our paper. In addition to the pointwise responses below, here we summarize our updates.

**[Extra Experiments.]**

*[Statistical Significance.]* As mentioned by reviewer **EKmy**, **qsko**, and **DuzK**, we carry out repeat experiments of Transformer-XL on SST-2, the results demonstrate our SMoE-Dropout achieves superior performance with no overlap of error bars, compared with other SMoE-variants and the dense counterpart.

*[Interesting Extension.]* Thanks to the interesting question from reviewer **Duzk**, we also evaluate our method on a multi-hop question answering task, i.e., HotpotQA. The results validate the effectiveness of our SMoE-Dropout on improving compositional generalization.

*[More Ablation.]* Additionally, we conduct further experiments including ablation on the number of total experts and applying SMoE-Dropout to both MLP and attention blocks (@reviewer **EKmy**); comparison against SMoE with diversity regularization (@reviewer **3ZBm**); ablation study about different schedules for progressively activating experts (@reviewer **qsko**); and applying curriculum learning to learning SMoEs (@reviewer **Duzk**).

**[Reproducibility.]**

- The implementation codes have been provided in the supplementary with training scripts, for the experiments in Table 1,2, and Figure 5 (@reviewer EKmy).

**[Paper Editing.]**

- Extra experiments are provided in Appendix A2.

- Details about Concrete Dropout and Dropblock are provided in Appendix A1.

- The abstract has been revised, in which we clarify the main focus of our SMoE-Dropout is to provide a new approach for training large-scale dense networks, rather than a better training recipe for SMoEs.

- All the mentioned types have been addressed.


We hope our pointwise responses below could clarify all reviewers’ confusion. Please kindly let us know if you have any further questions: we will be more than happy to address them fully.

Thanks again for all the reviewers' time.

---

### Decision · Program_Chairs · 2023-01-20

**Decision:**

Accept: notable-top-25%

**Justification For Why Not Higher Score:**

Empirical improvements are moderate. No theoretical justification for the approach.

**Justification For Why Not Lower Score:**

Efficient training/inference of/with large models is an increasingly relevant topic and sparse MoEs are an important direction for this.

**Metareview: Summary, Strengths And Weaknesses:**

The paper addresses the issues of representation collapse and scalability in sparse MoE transformer models. It proposes SMoE-Dropout that uses randomly initialized and fixed router network to address these issues. Empirical improvements are moderate but consistent. All reviewers appreciate the novelty of the idea and contributions of the paper, and the author response has addressed the reviewers' concerns well.


**Note From Pc:**

if the above contains the word "oral" or "spotlight" please see: "oral" presentation means -> notable-top-5% and "spotlight" means -> notable-top-25%. As stated in our emails, we are disassociating presentation type from AC recommendations